# The Bisphenol A Induced Oxidative Stress in Non-Alcoholic Fatty Liver Disease Male Patients: A Clinical Strategy to Antagonize the Progression of the Disease

**DOI:** 10.3390/ijerph17103369

**Published:** 2020-05-12

**Authors:** Alessandro Federico, Marcello Dallio, Antonietta Gerarda Gravina, Nadia Diano, Sonia Errico, Mario Masarone, Mario Romeo, Concetta Tuccillo, Paola Stiuso, Filomena Morisco, Marcello Persico, Carmelina Loguercio

**Affiliations:** 1Department of Precision Medicine, University of Campania Luigi Vanvitelli, via Pansini 5, 80131 Naples, Italy; alessandro.federico@unicampania.it (A.F.); antoniettagerarda.gravina@unicampania.it (A.G.G.); marioromeo@virgilio.it (M.R.); concetta.tuccillo@unicampania.it (C.T.); paola.stiuso@unicampania.it (P.S.); carmelina.loguercio@unicampania.it (C.L.); 2Department of Experimental Medicine, University of Campania Luigi Vanvitelli, Via De Crecchio 7, 80138 Naples, Italy; nadia.diano@unicampania.it (N.D.); sonia.errico@unicampania.it (S.E.); 3Department of Medicine and Surgery, Scuola Medica Salernitana, University of Salerno, Via Allende, Baronissi, 84081 Salerno, Italy; mmasarone@unisa.it (M.M.); mpersico@unisa.it (M.P.); 4Department of Clinical Medicine and Surgery, University of Naples Federico II, 80131 Naples, Italy; filomena.morisco@unina.it

**Keywords:** non-alcoholic fatty liver disease, obesity, cancer biology, liver

## Abstract

*Introduction*: Bisphenol A (BPA) exposure has been correlated to non-alcoholic fatty liver disease (NAFLD) development and progression. We investigated, in a clinical model, the effects of the administration of 303 mg of silybin phospholipids complex, 10 μg of vitamin D, and 15 mg of vitamin E (RealSIL, 100D, IBI-Lorenzini, Aprilia, Italy) in male NAFLD patients exposed to BPA on metabolic, hormonal, and oxidative stress-related parameters. *Methods*: We enrolled 32 male patients with histologic diagnosis of NAFLD and treated them with Realsil 100D twice a day for six months. We performed at baseline clinical, biochemical, and food consumption assessments as well as the evaluation of physical exercise, thiobarbituric acid reactive substances (TBARS), plasmatic and urinary BPA and estrogen levels. The results obtained were compared with those of healthy control subjects and, in the NAFLD group, between baseline and the end of treatment. *Results*: A direct proportionality between TBARS levels and BPA exposure was shown (*p* < 0.0001). The therapy determined a reduction of TBARS levels (*p* = 0.011), an improvement of alanine aminotransferase, aspartate aminotransferase, insulinemia, homeostatic model assessment insulin resistance, C reactive protein, tumor necrosis factor alpha (*p* < 0.05), an increase of conjugated BPA urine amount, and a reduction of its free form (*p* < 0.0001; *p* = 0.0002). Moreover, the therapy caused an increase of plasmatic levels of the native form of estrogens (*p* = 0.03). *Conclusions*: We highlighted the potential role of BPA in estrogen oxidation and oxidative stress in NAFLD patients. The use of Realsil 100D could contribute to fast BPA detoxification and to improve cellular antioxidant power, defending the integrity of biological estrogen-dependent pathways.

## 1. Introduction

In the near future non-alcoholic fatty liver disease (NAFLD) will represent the most important hepatopathy, the first cause of hepatocellular carcinoma (HCC) and liver transplantation, as clearly and widely documented in the scientific literature [1]. Over time, researchers worldwide have focused attention on the triggering factors for this pathology, affirming an important role played by the environment and genetics in its appearance. NAFLD belongs to a complex clinical context identified as metabolic syndrome, implying multisystemic derangements that make this pathology a challenge from a prognostic and therapeutic point of view [2]. The role of the environment in this scenario has been well explored, in particular regarding several pollutants able to intervene in the pathogenetic cascade, carrying out an important role for the comprehension of molecular mechanisms that support the disease. 

Particular attention has been focused on endocrine disrupting chemicals (EDCs), heterogeneous group of substances that interfere with the normal function of certain hormones (especially sexual and thyroid ones), particularly concentrated in some plastics and food containers produced on large scale and with large market distribution [3]. 

One of the most important EDCs is bisphenol A (BPA) and its human exposure is almost ubiquitarian because of high concentrations in preserved food, water, electronic components, compact disks, reused paper, etc. [4]. BPA exposure has been correlated to both obesity and hormone mediated insulin resistance through the phosphorylation of extracellular signal-regulated kinases (Erk-1), demonstrating a clear link with metabolic diseases [5]. In this field, our group recently demonstrated that the exposure to high levels of BPA was related to the worsening of NAFLD, giving a new pathogenetic theory in which environmental chemical substances, together with a hypercaloric and/or unhealthy diet, could trigger NAFLD progression [6]. 

In particular, we reported higher concentrations of BPA in plasma and urine of patients with NAFLD compared to healthy controls and, among NAFLD subjects, higher levels in those affected by non-alcoholic steatohepatitis (NASH) compared to simple steatosis (NAFL) ones [6]. 

The main excretion way for BPA elimination is represented through urine: specifically, BPA is continuously released in the blood by adipose tissue reservoir and eliminated by glomerular filtration. In a previous clinical model, the possibility to reduce the BPA plasmatic amount following a month BPA-free diet was highlighted, even if the effect on its urine amount was not significant [6]. Finally, a direct correlation between the plasmatic concentration of BPA and the grade of liver inflammation, in accordance with Kleiner et al. criteria, was demonstrated [6]. 

In another in vitro study it was demonstrated that BPA was able to induce, in a dose- and time-dependent manner, a higher Human Hepatocellular Carcinoma Cell line (HepG2) proliferation rate in comparison to control cell cultures, inducing, moreover, an increase of thiobarbituric acid reactive substances (TBARS), used as markers of lipid peroxidation and oxidative stress [7,8]. The highlighted effects could be due to an increase of protein synthesis directly involved in the regulation of the cell cycle, such as Erk, phosphorylated-Erk and protein kinase B (AKT). 

In this setting BPA seems to be able to increase both aryl hydrocarbon receptor (Ahr) levels and its translocation into the nucleus, inducing the transcription of genes involved in the cell inflammation and proliferation [7,8,9].

BPA has been demonstrated to enhance estrogen production in cell cultures, particularly the oxidized form of these hormones. All these effects have been antagonized by the concomitant introduction of silybin, a derivative of *carduus marianus,* in cell culture. This herbal compound is able to interfere with all the aforementioned biological events, demonstrating a direct counter-effect to the pathological mechanisms carried out by BPA [7]. 

On the basis of this scientific evidence the aim of this study was to test in a clinical model, the effects of silybin with phospholipids and vitamin D (RealSIL, 100D, IBI-Lorenzini, Aprilia, Italy) administration in NAFLD patients exposed to BPA on metabolic, hormonal, and oxidative stress-related parameters.

## 2. Materials and Methods 

### 2.1. Patients

This prospective study is in compliance with ethical guidelines of the Declaration of Helsinki (1975) and was approved by the ethical committee of the University of Campania Luigi Vanvitelli in Naples (protocol n. 531/2016). Thirty-two consecutive male patients with histological diagnosis of NAFLD and as control group, 30 patients without NAFLD, followed at the Hepatogastroenterology Divisions of the University of Campania Luigi Vanvitelli, were enrolled between January and October 2017. Inclusion criteria were as follows—age between 18 and 80, male sex, and NAFLD diagnosis—whereas exclusion criteria were: female sex, type 2 diabetes mellitus, use of hepatoprotective drugs, presence of tumors or chronic inflammatory disease such as inflammatory bowel disease, rheumatoid arthritis, systemic lupus erythematosus or other major systemic diseases, ongoing infections, acute or chronic kidney disease, alcohol or drug abuse history, cirrhosis or other causes of chronic liver damage, and psychological/psychiatric problems that could invalidate the informed consent. 

We decided not to include the female gender in the study because of the significant interpretative difficulties related to the analysis of the outcomes of the hormonal estrogenic assessment. 

The definition of the presence/absence of NAFLD and the staging of the disease were diagnosed after the exclusion of other causes of liver diseases, by serological tests, clinical data, and by performing a liver biopsy. We also performed, in both NAFLD and control group, an abdominal ultrasound and a Fibroscan controlled attenuation parameter (CAP) evaluation.

Medical history, alcohol consumption, drug intake, current drug treatments, smoking habits, and blood pressure were investigated. Weight, height, and waist-to-height ratio (WHtR) were directly measured by using standardized devices. Body mass index (BMI) was also calculated by dividing the weight (kg) by height in meters squared. 

Additional data included routine laboratory tests (blood glucose and insulin, homeostasis model assessment (HOMA-IR), aspartate and alanine aminotransferases, gamma-glutamyl transpeptidase, blood count, C-reactive protein, tumor necrosis factor alpha) were obtained by blood peripheral venous samples [10]. Blood samples were collected after ≥10 hours fast and rapidly processed or stored at −20 °C after centrifuging. 

A complete blood count was performed by an automated analyzer. Glucose, insulin, gamma-glutamyl transpeptidase, C-reactive protein, aspartate aminotransferase, and alanine aminotransferase levels were measured enzymatically using commercially available kits (R&D Systems, Minneapolis, MN, USA). Insulin sensitivity was calculated by Homeostatic model assessment (HOMA) using the formula:HOMA = fasting insulin (μU/mL) × plasma glucose (mmol/L) / 22.5.

Sera were tested for tumor necrosis factor alpha by the enzyme-linked immunosorbent assay (ELISA), according to the manufacturer’s instructions (Quantikine ELISA kit, R&D SYSTEMS). All the analyzed parameters were repeated, in NAFLD group, at baseline (T0) evaluation and at the end of therapy (T6).

### 2.2. Nutritional Assessment

Food intake was evaluated by software analysis, Winfood Software 2.0 package (Medimatica s.r.l., Martinsicuro, Italy), in all enrolled subjects. On the basis of the quantity and quality of foods consumed, the program evaluates the energy intake and the percentage of macronutrients and micronutrients in each food. The complete elaboration of intakes shows the list of diet components, the ratio among components, the calories, and the subdivision into breakfast, lunch, and dinner. 

We recorded the food intake of a complete week, including working days and the weekend. Data were compared with the tables of food consumption and recommended dietary intakes of the Italian National Institute of Nutrition and Food Composition Database in Italy and alcohol use was evaluated with a standardized pre-codified questionnaire (complete AUDIT test) [11,12]. 

The quantity of daily alcohol intake was calculated based on a “drink” that corresponds to about 12 g of pure ethanol [13]. Moreover, we assessed the consumption of BPA enriched foods with a one-month evaluation questionnaire [10]. 

### 2.3. TBARS, BPA, and Hormonal Assessment

We evaluated TBARS as a marker of oxidative stress in patients’ sera [14]. The assay was performed with 10 μL of serum and the chromogen TBARS was quantified using a spectrophotometer at a wavelength of 532 nm with 1,1,3,3-tetramethoxypropane as standard. The amount of TBARS was expressed as nmol/μg of protein and we presented the data as mean (M) ± standard deviation (SD) [10]. Plasma and urine samples were also collected from NAFLD patients and controls, then stored frozen in glass vials at −20 °C until analysis. 

The urine samples (3 mL collected by the 24 h collection of urine) were prepared to measure the levels of conjugated and free BPA following the previously validated method [14,15]. The measured BPA concentration was corrected for creatinine values, moreover we calculated the rate of excretion per hour of BPA using the formula: (total urine concentration in ng/mL × 24 h urine volume in mL)/24.

BPA and 17-β-Estradiol (E2) were simultaneously extracted from plasma samples using methanol as organic solvent (liquid/liquid extraction) and purified by solid phase extraction (SPE). 

The procedure described by Nicolucci et al. (2017) was modified using specific cartridges for steroids [14]. LC-MS/MS analysis was performed using a Dionex UltiMate3000 High-performance liquid chromatography (HPLC) system (Thermo Fisher Scientific Inc, Monza, Italy), coupled to a triple quadrupole mass spectrometer by an electrospray ion source, switching in negative ion mode for 17-β-estradiol and BPA and positive mode for testosterone [16,17]. 

The separation of target compounds was achieved by a Phenomenex Kinetex F5 reversed phase column (100 × 4.6 mm, 2.6 μm). Chromatography was run at room temperature by linear gradient elution; water and methanol were used as mobile phases, contained 0.1% acetic acid (v/v) [15,16]. The analytes were quantified in multiple reaction monitoring (MRM) mode. The following ion transitions were monitored: m/z 227.1 → m/z 212.1 (quantifier) and m/z 227.1 → m/z133.2 (qualifier) for BPA; m/z 271.0 → m/z 145.0 (quantifier) and 271.0 → m/z 183.0 (qualifier) for E2; m/z 288.9→ m/z 97.1 (quantifier) and m/z 288.9 → m/z 109.2 (qualifier) for testosterone. The linearity of the detector response was verified over the concentration range 0.100–200 ng/mL for each analyte. 

We expressed the results as mean ± SD.

### 2.4. Experimental Design

All 32 NAFLD enrolled patients had oral administration of 303 mg of silybin-phospholipids complex, 10 μg of vitamin D, and 15 mg of vitamin E, twice a day for six months. The amount of vitamin E used in this pharmacological complex is too low to consider it as an active component of therapeutic effect in this setting. On the contrary it is essential from a drug molecular design point of view, to give to the drug complex chemical stability.

During the study period the patients were on free diet on the basis of dietary habits prior to the enrollment, therefore, it was advised there were no restrictions on the use of BPA products. Moreover, we performed at baseline clinical, biochemical, fibroscan CAP, abdominal ultrasound, food consumption assessments and the evaluation of physical exercise, TBARS, estrogens, plasmatic and urinary BPA levels measurement. We performed a baseline comparison of these parameters between the two study groups in order to highlight the major differences.

At the end of treatment period we evaluated the effect of therapy on TBARS, several biochemical indices, plasmatic and urinary BPA and estrogen levels in the NAFLD group (Figure 1).

### 2.5. Sample Size and Statistical Analysis

The study that validated the BPA urinary level quantization method, identified as reference values 0.48 ± 0.16 ng/mL in normal weight young subjects and 0.79 ± 0.16 in obese young subjects [16].

On the basis of this difference, also considering a variability of 10% as compared to the given values, we estimated that 60 patients (30 for each arm of the study) was the correct sample size number of subjects to be investigate for a significant difference in BPA levels, with an 0.01 alpha error and a 90% statistical power in a two-sided test with a 95% Confidence Interval. Wilcoxon signed ranks test and Mann–Whitney U test were performed in order to compare continuous variables.

A Kolgoromov–Smirnov for normality was performed to evaluate if parametric or non-parametric analysis should be applied. Statistical significance was defined as *p* < 0.05 in a two-tailed test with a 95% Confidence Interval.

All continuous variables were expressed as mean ± standard deviation/standard error.

We performed a multivariate analysis in order to evaluate the role of some confounding variables on the comparison between serum TBARS and BPA levels in NAFLD patients. To do that we used as independent variable serum BPA levels and as dependent ones: TBARS, age, BMI, waist/hip ratio, comorbidity (obesity, hypertension, compensated cardiovascular diseases), and histologic liver staging.

Statistical analyses were performed using the Statistical Program for Social Sciences (SPSS^®^) vs.18.0 (IBM, Armonk, NY, USA).

## 3. Results

### 3.1. Clinical and Biochemical Assessment

The main features of the considered population are shown in Table 1.

The enrolled patients had a NAFLD activity score between 4 and 8 and no cirrhotic patients were enrolled. All NAFLD patients demonstrated a CAP value >250 dB/m and a stiffness value <13 kPa. On the contrary, controls showed CAP values <250 dB/m and stiffness <5.5 kPa.

NAFLD patients demonstrated no difference in terms of food habits in comparison to healthy ones in terms of total quantitative of daily calories: 2705 ± 252 controls vs 2782 ± 188 NAFLD patients (*p* = 0.556).

On the contrary we found some differences in terms of quality of the daily intake of macro- and micronutrients. In particular, NAFLD patients showed a higher intake of proteins, glucose, fructose, sucrose, maltose, saturated fatty acid (SFA), monounsaturated fatty acid (MUFA), folic acid, vitamins A and C, and thiamine in comparison to controls (*p* < 0.001).

On the other hand, the daily intake of polyunsaturated fatty acid (PUFA), riboflavin, and vitamin B6 was lower in NAFLD group (*p* < 0.001).

Although NAFLD patients did not show significant differences in terms of daily caloric intake they mainly used food contained in non BPA-free plastic containers, as well as fast food products during the study period, as highlighted by submission of a specifically prepared evaluation test: 5 ± 2 meals per month in controls vs 13 ± 7 meals per month in NAFLD patients (*p* = 0.021).

We reported higher BMI and waist/hip ratio values in NAFLD patients in comparison to controls (both *p* < 0.001), moreover a higher prevalence of cardiovascular pathologies (*p* = 0.032) was also found.

Other significant differences between NAFLD and the control group were found in insulinemia and homeostasis model assessment (HOMA) which were higher in NAFLD patients (*p* = 0.004 and *p* = 0.018, respectively). Finally, NAFLD patients showed higher levels of alanine aminotransferase (ALT), TNF-α, plasmatic, free urine amount, and total urine amount of BPA but not of aspartate aminotransferase (AST) in comparison to controls (*p* < 0.001, *p* = 0.002, *p* < 0.001, *p* < 0.001, *p* < 0.001, *p* = 0.913 respectively). Moreover, C reactive protein (CRP) was higher in the NAFLD group in comparison to control (*p* < 0.001).

During the period of the study patients were on free diet on the basis of dietary habits prior to the enrollment, therefore, no restrictions were advised about the use of BPA-containing products.

Moreover, we also performed the food consumption assessment and the evaluation of physical exercise at the end of the treatment period, without finding significative differences in these parameters in comparison to the baseline.

We observed a statistically significant improvement of several biochemical indices evaluated in the NAFLD group at the end of treatment in comparison to the baseline: ALT, insulinemia, HOMA-IR, CRP, tumor necrosis factor (TNF)-α (Figure 2) (*p* < 0.05).

### 3.2. Oxidative Stress Assessment

NAFLD patients showed an increase in the evaluated marker of oxidative stress. We demonstrated, in a previous study, that the HepG2 cells exposure to BPA (0.05 LM) for 48 h was able to induce a significant increase of TBARS levels in comparison to unexposed cells (*p* < 0.0001), highlighting its potential role to induce lipid peroxidation [6].

Accordingly, in the present clinical setting, TBARS was found to be higher in NAFLD patients compared to controls (*p* = 0.013). In particular, assuming that TBARS is an unspecific marker of oxidative stress and lipid peroxidation, we compared plasma TBARS and BPA of patients with NAFLD, as can been seen in Figure 3, highlighting a direct proportionality between them independently of other evaluated variables: BMI, waist/hip ratio, comorbidity (obesity, hypertension, compensated cardiovascular diseases), and histologic liver staging (*p* < 0.0001) (Figure 3).

Even if there was a continuous exposure of NAFLD patients to BPA, as confirmed by non-significant variation of plasmatic and urinary levels at T6 in comparison to the baseline (*p* = 0.732 and *p* = 0.915, respectively), the administration of 303 mg of silybin-phospholipids complex, 10 μg of vitamin D, and 15 mg of vitamin E led to a significant reduction of TBARS levels at the end of treatment (*p* = 0.011) (Figure 4). At the same time the therapy determined in NAFLD patients a statistically significant increase of conjugated BPA (CBPA) urine amount concomitantly with a reduction of its free form (*p* < 0.0001, *p* = 0.0002 respectively) (Figure 5). Moreover, we observed an increase of BPA excretion rate per hour at the end of therapy in comparison to the baseline (T0: 373.1 ± 240.3 ng/mL/h; T6: 465.7 ± 293.1 ng/mL/h; *p* = 0.019), without a statistically significant modification of daily urine volume or total daily BPA urine amount.

### 3.3. Steroid Hormones

In our clinical setting we demonstrated that estrogen levels were higher in the NAFLD population in comparison to controls (*p* = 0.023). The 303 mg of silybin-phospholipids complex, 10 μg of vitamin D, and 15 mg of vitamin E administration caused an increase in plasmatic levels of estrogens in their native and so biological active form: T0 = 0.74 ± 0.71 (ng/mL); T6 = 4.57 ± 10.29 (ng/mL) (*p* = 0.03).

## 4. Discussion

In the era of metabolic liver diseases, the knowledge of the pathogenetic mechanisms supporting NAFLD and causing its progression to more advanced stages, represents the key point for the comprehension of prognosis and for the choice of the better therapy.

The attention paid on this topic by the scientific community has led to different pathogenetic theories to be realized that take into consideration the role of the environment in NAFLD development [18]. The analysis of molecular mechanisms supported by environmental contaminants, like BPA, in metabolic pathologies such as overweight, obesity, insulin resistance, diabetes, and NAFLD is part of this context and a well explored fascinating field of the current scientific literature [19,20]. NAFLD usually is characterized by the maintenance of a low-grade inflammatory state, in particular in the case of concomitant diagnosis of metabolic syndrome. Therefore, the research regarding NAFLD treatment has been partially focused on the use of anti-inflammatory and anti-oxidant therapies, such as silybin, obtaining different outcomes [21,22,23].

In our study we excluded from the enrollment patients with a diagnosis of diabetes mellitus. Since BPA exposition can induce diabetes mellitus, we made this choice in order to avoid the effect of this disease on the study results, showing the consequence of BPA exposition prior to the development of diabetes mellitus [24].

During the study period, the patients were not instructed to maintain a particular dietary regimen or increase daily physical exercises, as confirmed by the quantitative analysis of these parameters at the end of the treatment.

Moreover, we did not advise any behavioral change to avoid exposure to BPA, in order to keep stable the plasmatic and urinary BPA concentration during the study period. In fact, the concentration of total BPA in plasma and urine of NAFLD patients did not modify in a statistically significant way after 6 months of treatment if compared to the baseline.

As regards food behavior, NAFLD patients did not show a higher significant daily caloric consumption in comparison to controls.

This observation suggests that the link between NAFLD and hypercaloric diet could also be scientifically re-explored in the light of the possibility of an underreporting of the food record made by obese subjects [25,26]. On the other hand, significant differences in daily assumed micronutrients were recorded and are in line with other similar results from the scientific literature [27]. Moreover, NAFLD patients more often assumed fast-food or large-scale distributed precooked food if compared to controls. We would like to underline that not only in regard to their caloric amount, the potential effect of this food category in inducing NAFLD, may also be related to high levels of BPA contained in the packaging, as confirmed by higher levels of BPA plasma, free urine and total urine concertation of NAFLD patients in comparison to controls.

Another characteristic of NAFLD patients was represented by a statistically significant higher BMI value, as well as WHtR.

Although this is not surprising, because NAFLD is mainly prevalent in obese patients [28,29], it is important to consider the effect of adipose tissue on the endocrine metabolism. In fact, one of the most important roles is represented by the activity of aromatase which causes testosterone conversion to estradiol [30]. In this context the adipose tissue is considered as an organ characterized by a complex functionality, being one of the main actors in the pathogenesis of metabolic disorders [31,32]. Adipocyte dysfunction would underlie a multisystemic altered response that leads to trigger NAFLD and allow its progression to more advanced stages. The topic is even more interesting if we consider adipose tissue as the main site of BPA bioaccumulation and release in the blood, determining a continuous tissue exposure to this environmental contaminant [6,33].

Despite the fact that the enrolled patients were not affected by diabetes, they demonstrated both higher HOMA and insulinemia values if compared to controls. This result needs more attention, considering the close connection between NAFLD and insulin resistance. In fact, we demonstrated, in a previous in vitro study, that the main harmful action related to BPA exposure in terms of induction of oxidative stress, lipoperoxidation, and cell proliferation in HepG2 cell cultures was particularly significant in the presence of high concentrations of glucose (H-HepG2) [7].

This methodology used to simulate a condition of hyperglycemia, a typical condition of diabetic patients, demonstrated that the higher harmful effect related to BPA exposure may occur in patients with type 2 diabetes mellitus. Therefore, the patients taken into consideration in our clinical setting, even if they did not have a clear diagnosis of diabetes mellitus, represent a high-risk BPA exposure population for the possibility to develop a type 2 diabetes mellitus. This is especially the case if we consider the outcomes of this study concerning the evaluation of parameters of oxidative stress, as successively discussed. In this regard, TBARS and CRP have been demonstrated to be higher in NAFLD patients in comparison to controls. These data are in accordance with the consideration of NAFLD as a pathology characterized by an increase of oxidative stress linked to low grade inflammation [21,34].

The treatment with 303 mg of silybin-phospholipids complex, 10 μg of vitamin D, and 15 mg of vitamin E determined a statistically significant improvement of several biochemical markers: AST, ALT, insulinemia, HOMA-IR, CRP, and TNF-α, confirming its effect as an antioxidant/anti-inflammatory drug, potentially useful in this clinical context.

Moreover, in order to understand the relationship between BPA exposure and oxidative stress, we analyzed the type of proportionality between BPA and TBARS plasma levels. We demonstrated that for higher BPA exposures, higher plasma TBARS levels were found, independently from other parameters evaluated in the study such as BMI, waist/hip ratio, comorbidity, and disease stage. Amounts of 303 mg of silybin-phospholipids complex, 10 μg of vitamin D, and 15 mg of vitamin E administration determined a reduction of plasma TBARS levels, despite continuous BPA exposure. From these outcomes and the analysis of scientific literature, it is possible to conclude that NAFLD is characterized by an increase of inflammatory state and oxidative stress status. This assumption is very important in the light of the possibility to generate an inflammatory tissue microenvironment, that in turn could be responsible for an alteration of metabolic pathways involved in the synthesis of proteins or hormones able to stop or slow down NAFLD.

The fact that BPA belongs to the EDC category led us to study the effect on estrogen synthesis and function of HepG2 cell exposure [7].

Particularly BPA could be able to cause an increase of steroid hormones synthesis, especially estrogens (Figure 6).

The study of biological functions related to estrogen signaling represents one of the most important topics in the chronic liver disease context. In a previous study we highlighted the increase of oxidized estrogens in HepG2 cell cultures exposed to BPA, opening a way for a new point of view in the comprehension of HCC pathogenesis [7].

In this context, we found higher plasma estradiol levels in NAFLD patients in comparison to controls.

The reason could be mainly linked to two mechanisms: first, direct effect of BPA in the biosynthesis of steroid hormones through a pathway that involves an aryl hydrocarbon receptor, the translocator protein (18 kDa) (TSPO), and the steroidogenic acute regulatory protein (STAR) [7,35], second, the biological activity of aromatase, particularly in our study population due to higher BMI and waist/hip ratio compared to controls.

However, the main estrogen biosynthesis occurs in a tissue microenvironment that supports its fast oxidation. The role of estrogens in NAFLD and HCC has been studied over time by many research groups. Growing evidence has demonstrated a protective effect linked to estrogen biosynthesis towards metabolic and cardiovascular pathologies [36,37].

They regulate lipid metabolism and would underlie the protection of fissuring of atherosclerosis plaque [36,37]. On the other hand, estrogenic activity could be also involved against leptin-induced HCC pathogenesis, related to the high production of leptin in obese patients. Estrogens could be able to modulate the activity of several enzymes involved in HCC appearance: signal transducer and activator of transcription (STAT) protein family, suppressors of cytokine signaling (SOCS) proteins, ERK, and G protein-coupled estrogen receptor 1 (GPER) proteins [38,39].

The main biosynthesis of estrogens, which quickly reaches oxidation, as demonstrated in our previous work, would expose these patients to an increase of HCC development due to the loss of protective effects that estrogens could exert in this context [7].

The 303 mg of silybin-phospholipids complex, 10 μg of vitamin D, and 15 mg of vitamin E administration for six months led to a statistically significant increase of circulating estrogen levels in their native form. It is possible to suppose that, in accordance with the antioxidant potential possessed by silybin and confirmed by a reduction of plasma TBARS levels, the increased amount of estrogens could be due to the reduction of the oxidized form of the hormones, normally not detected using the methodology described above. The treatment with 303 mg of silybin-phospholipids complex, 10 μg of vitamin D, and 15 mg of vitamin E determined a statistically significative increase of CBPA levels in urine. These data are interesting because a BPA-free diet is also able to reduce BPA circulating levels, but it takes a longer time.

The conversion from BPA to CBPA represents an important step in the metabolism of this environmental contaminant because, first it nullifies the biological harmful BPA effects and, second, it creates a conjugated compound eliminated fast through kidney filtration as confirmed by the increase of the rate per hour excretion. The next aim is to clarify the relationship between Realsil 100D therapy and the risk of NAFLD worsening/complications also bearing in mind the effect of the therapy on estrogen metabolism of female patients.

## 5. Conclusions

Since BPA exposition could be responsible for the worsening of NAFLD, the aim of this study was oriented to evaluate the effect derived from silybin with phospholipids and vitamin D (RealSIL 100D) administration in NAFLD patients exposed to BPA on metabolic, hormonal, and oxidative stress-related parameters. BPA was able to reduce the TBARS amount, used as marker of lipoperoxidation, to improve several biochemical indices such as insulinemia, HOMA-IR, ALT, CRP, and TNF-α, as well as to increase the circulating estrogen levels in their non-oxidized form, consenting them to maintain their biological functions. Moreover, despite a BPA-free diet being able to reduce tissue exposure to this environmental contaminant in a statistically significant way, the use of silybin seems to be able to increase the CBPA urine excretion rate per hour, enhancing its elimination. The use of silybin seems to be promising due to the demonstrated effect on different worsening factors involved in this complex pathological context.

## Figures and Tables

**Figure 1 ijerph-17-03369-f001:**
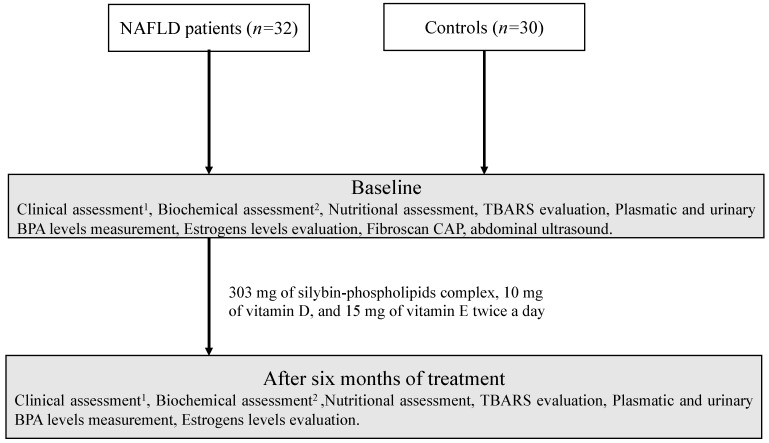
Study design flow-chart. ^1^ Weight, height, and waist-to-height ratio, body mass index, blood pressure measurement; ^2^ Blood glucose and insulin, homeostasis model assessment, aspartate and alanine aminotransferases, gamma-glutamyl transpeptidase, blood count, C-reactive protein, tumor necrosis factor alpha. NAFLD: non-alcoholic fatty liver disease; TBARS: thiobarbituric acid reactive substances; BPA: bisphenol A; CAP: controlled attenuation parameter.

**Figure 2 ijerph-17-03369-f002:**
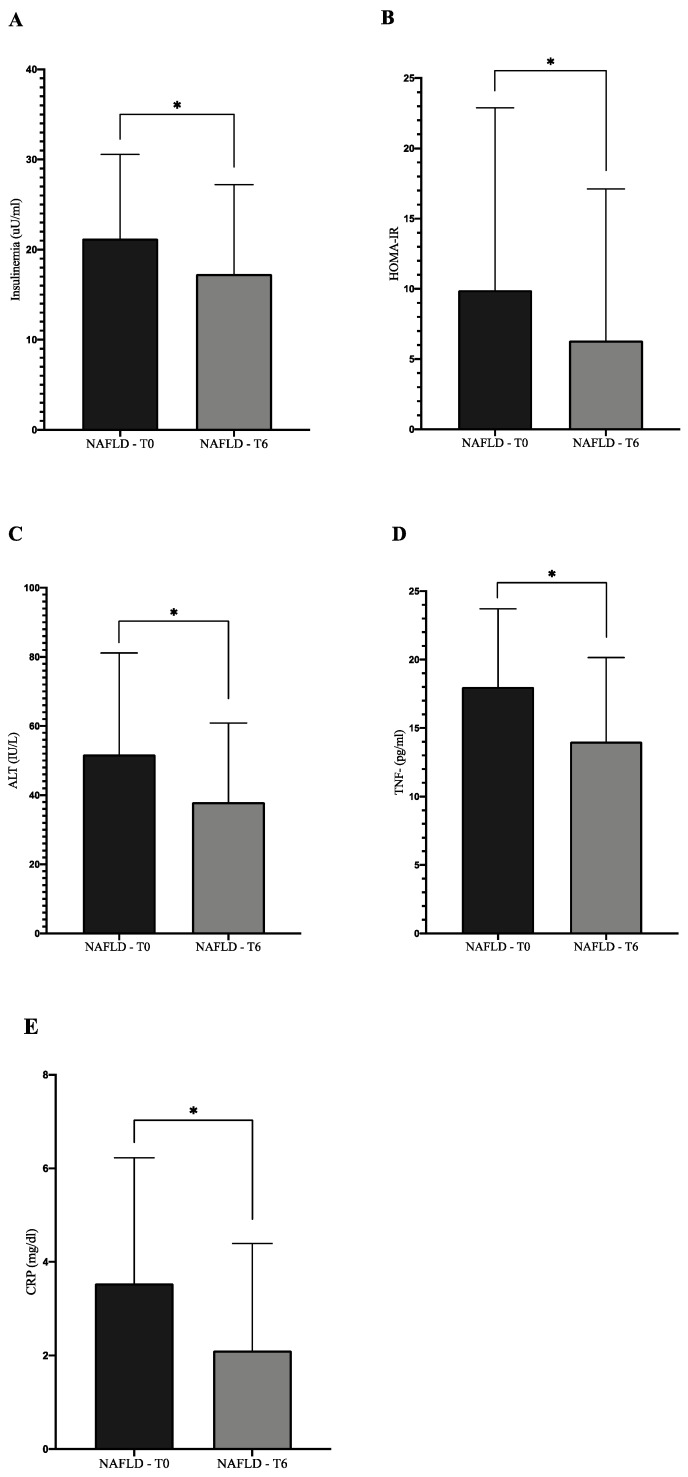
Effect of Realsil 100D administration on several biochemical indices in NAFLD patients. All the evaluated parameters showed a statistically significant improvement at the end of therapy in comparison to the baseline (* *p* < 0.05). NAFLD: non-alcoholic fatty liver disease; HOMA-IR: Homeostatic Model Assessment of Insulin Resistance; ALT: alanine aminotransferase; TNF-α: tumor necrosis factor alpha; CRP: C reactive protein. (**A**) Insulinemia, (**B**) HOMA-IR, (**C**) ALT, (**D**) TNF, (**E**) CRP.

**Figure 3 ijerph-17-03369-f003:**
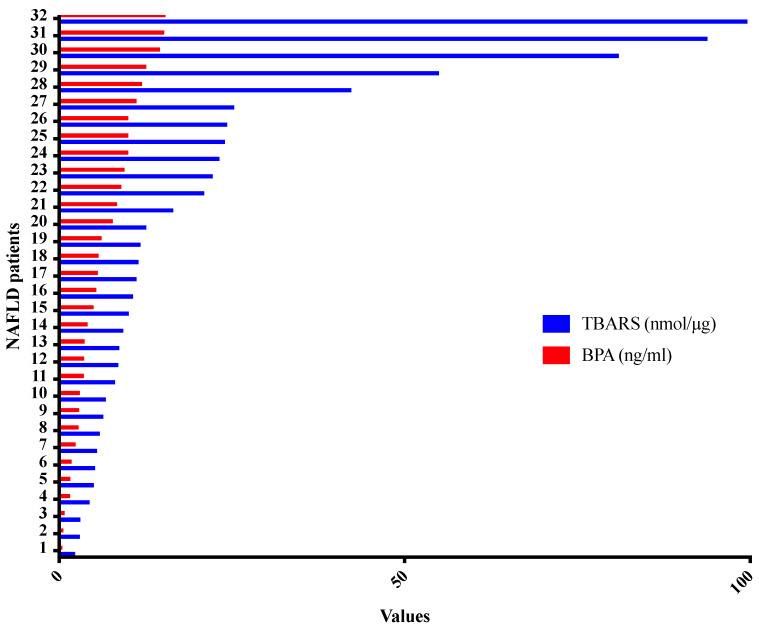
Proportionality relationship between thiobarbituric acid reactive substances (TBARS) and serum bisphenol A (BPA) levels in non-alcoholic fatty liver disease patients (*p* < 0.0001).

**Figure 4 ijerph-17-03369-f004:**
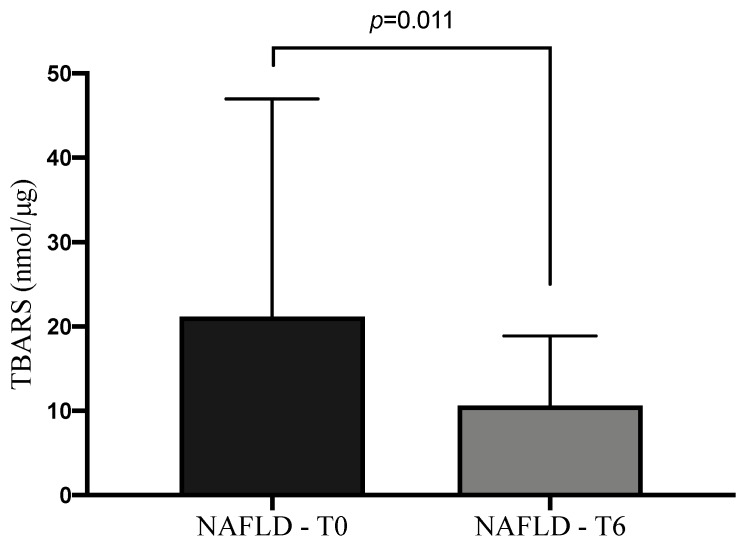
Thiobarbituric acid reactive substances (TBARS), levels at baseline and after six months of silybin with phospholipids and vitamin D (RealSIL 100D) administration (*p* = 0.011).

**Figure 5 ijerph-17-03369-f005:**
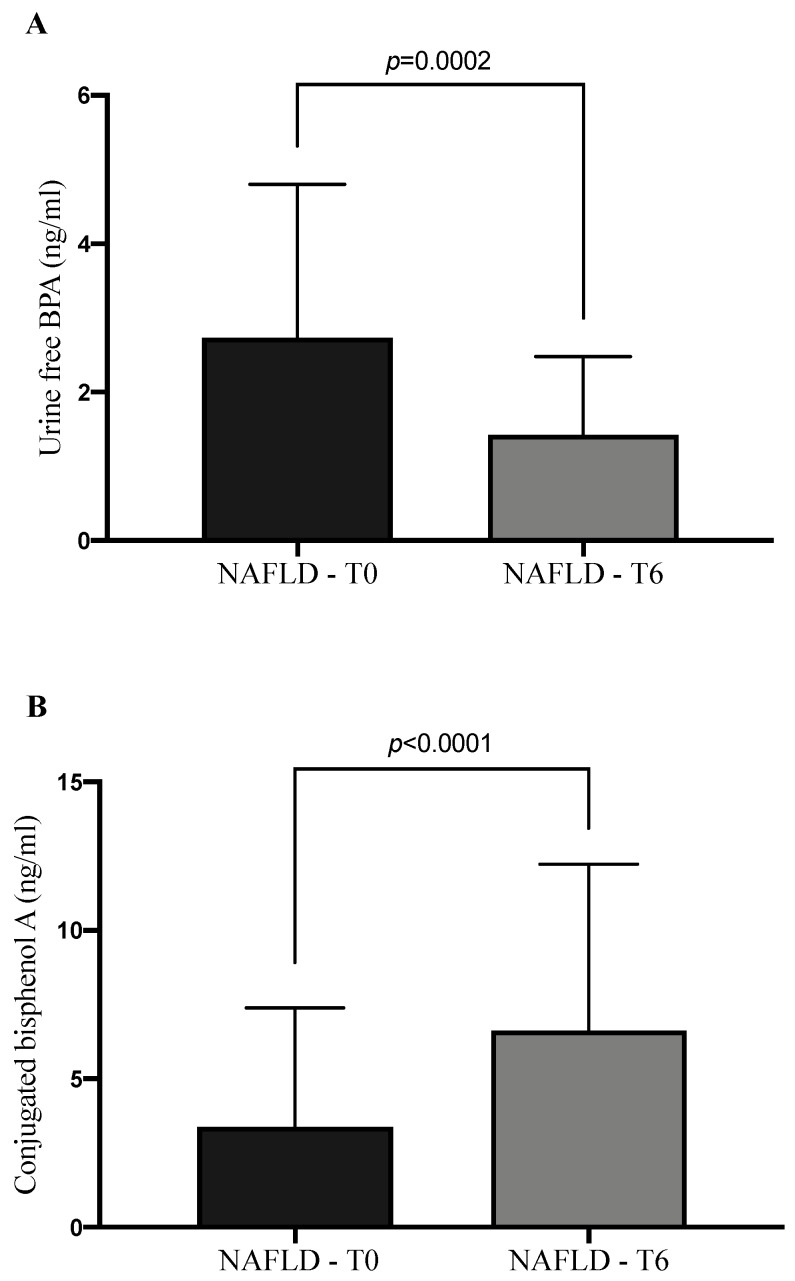
Free bisphenol A (BPA) and conjugated bisphenol A urine levels at the baseline and after six months of silybin with phospholipids and vitamin D (RealSIL 100D) administration ((**A**) *p* = 0.0002; (**B**) *p* < 0.0001 respectively).

**Figure 6 ijerph-17-03369-f006:**
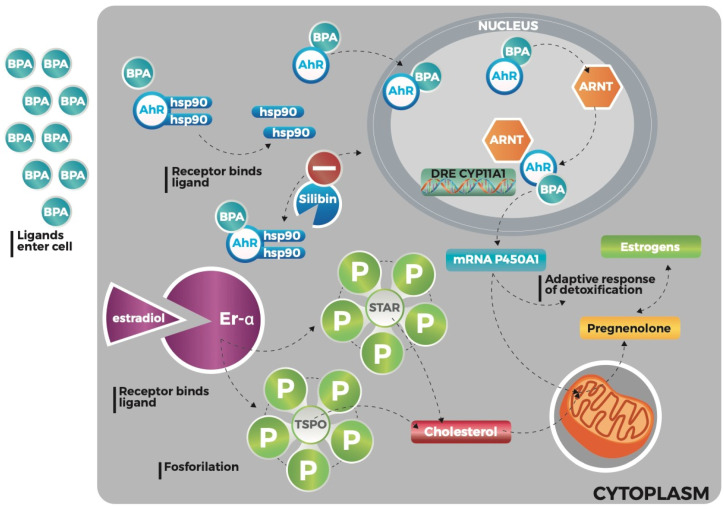
The bisphenol A-induced synthesis of steroids: The bisphenol A (BPA) spreads out through the membrane and binds the cytosolic multi-protein aryl hydrocarbon receptor (AhR), allowing its release in the cytoplasm by the heat shock protein (hsp)-90. Then the ligand-AhR complex dimerizes with a nuclear protein AhR Nuclear Translocator (ARNT): this process converts the AhR into the high affinity binding DNA form. The binding of the multimiginous complex ligand AhR-ARNT to its specific DNA recognition site stimulates the transcription of these genes with subsequent production of mRNA for cytochrome P450A1. Cytochrome P450A1 activates an adaptive detoxification response and stereogenesis by inducing the conversion of cholesterol into pregnenolone, which in turn will enter the steroid hormone synthesis pathway. In the meantime, the activation of estrogen receptor- alpha, due to the excess of estrogens for the aromatase functioning, induces the activation of steroidogenic acute regulatory protein (STAR) and translocator protein (18 kDa) (TSPO) that in turn enhance the transport of cholesterol across the mitochondrial membrane.

**Table 1 ijerph-17-03369-t001:** Baseline characteristics of enrolled patients (mean ± SD).

Variable	Controls *n* = 30	Study Population *n* = 32	*p*
Age (y)	44.5 ± 18.25	50.09 ± 12.38	0.217
BMI (kg/m^2^) ^1^	26.3 ± 3.39	30.26 ± 4.13	<0.001
Systolic blood pressure (mmHg)	121.6 ± 12.75	125.8 ± 12.68	0.13
Diastolic blood pressure (mmHg)	73.7 ± 8.15	76.81 ± 8.39	0.084
WHtR ^2^	0.84 ± 0.16	1.05 ± 0.25	<0.001
Fibroscan stiffness (kPa)	4.35 ± 0.65	5.5 ± 1.8	0.0119
Fibroscan CAP (dB/m)	193.1 ± 26.1	312.1 ± 32.2	<0.0001
Average daily energy intake (kcal)	2705 ± 252	2782 ± 188	0.556
Total daily proteins (% of total energy intake)	21.2 ± 10.4	27.3 ± 2.9	<0.001
Soluble carbohydrates (g/day)	82.7 ± 19.8	96.9 ± 16.9	<0.001
Saturated fatty acids (% of total energy intake)	7 ± 2.63	10.5 ± 1.9	<0.001
Monounsaturated fatty acids (% of total energy intake)	4.85 ± 0.9	13.5 ± 3.6	<0.001
Polyunsaturated fatty Acids (% of total energy intake)	23.7 ± 5.8	6.2 ± 3.3	<0.001
Folic acid (μg per day)	294 ± 91.7	378.6 ± 133.2	<0.001
Vitamin A (μg per day)	693 ± 179.8	871.9 ± 374.5	<0.001
Vitamin C (μg per day)	128 ± 47.6	161.7 ± 45	<0.001
Thiamine (μg per day)	0.8 ± 0.5	1.2 ± 0.5	<0.001
Riboflavin (μg per day)	4.2 ± 2.5	2.3 ± 1.3	<0.001
Vitamin B6 (μg per day)	1.9 ± 0.5	1.3 ± 0.3	<0.001
Red blood cells (×10^6^/mm^3^)	4.82 ± 0.65	4.81 ± 0.89	0.875
White blood cells (×10^3^/mm^3^)	4.75 ± 9	5.21 ± 1.55	0.455
Hemoglobin (g/dL)	13.57 ± 1.25	13.13 ± 1.12	0.153
PLT ^3^ (×10^3^/mm^3^)	208.2 ± 45.8	233.6 ± 69.5	0.167
FPG (mg/dL) ^4^	87.77 ± 16.95	95.19 ± 11.6	0.072
Insulinemia (μU/mL)	14.83 ± 5.66	21.23 ± 9.33	0.004
HOMA-IR ^5^	5.52 ± 8.57	9.91 ± 12.98	0.018
GGT (IU/L) ^6^	49.27 ± 33.92	63.44 ± 76.57	0.472
AST (IU/L )^7^	34.1 ± 20.15	38.53 ± 34.39	0.913
ALT (IU/L) ^8^	32.57 ± 13.5	51.75 ± 29.4	<0.001
TBARS (nmol/μg) ^9^	7.48 ± 2.9	21.2 ± 25.77	0.013
CRP (mg/dL) ^10^	1.48 ± 1.56	3.54 ± 2.68	<0.001
TNF-α (pg/mL) ^11^	13.56 ± 4.55	17.99 ± 5.71	0.002
Plasma BPA (ng/mL) ^12^	2.27 ± 1.49	6.45 ± 4.51	<0.001
Urine free BPA (ng/mL)	1.23 ± 1.09	2.73 ± 2.06	<0.001
Urine total BPA (ng/mL)	1.58 ± 1.23	5.84 ± 3.07	<0.001
Plasma beta estradiol (ng/mL)	0.29 ± 0.24	0.74 ± 0.71	0.023

^1^ BMI: body mass index; ^2^ WHtR: waist-to-height ratio; ^3^ PLT: platelets; ^4^ FPG: fasting plasma glucose; ^5^ HOMA-IR: homeostatic model assessment for insulin resistance; ^6^ GGT: gamma-glutamyl transferase; ^7^ AST: aspartate aminotransferase; ^8^ ALT: alanine aminotransferase; ^9^ TBARS: thiobarbituric acid reactive substances; ^10^ CRP: C-reactive protein; ^11^ TNF-α: tumor necrosis factor alpha; ^12^ BPA: bisphenol A.

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
