# Peer review of "The Bisphenol A Induced Oxidative Stress in Non-Alcoholic Fatty Liver Disease Male Patients: A Clinical Strategy to Antagonize the Progression of the Disease"

_ijerph, 2020, doi:10.3390/ijerph17103369_

Round 1

Reviewer 1 Report

The work of Frederico A and colleagues aim to discuss the effects bisphenol A (BPA) exposure in the Nafld pathology and the supplementation of antioxidants and vit d in this context.

I have some comments and suggestion that may help improving the manuscript.

It is stated in the abstract that the effects of a supplement were investigated in Nafld patients exposed to BPA, however, in the conclusion it is stated that a treatment was not one of the main objectives of the study. The same can be observed in the discussion. Authors must clearly stat the hypothesis of the study, main objective, the finds that support it and a conclusion that answers the main objective of the study.

How these parameters were analyzed?

  • Blood glucose and insulin, aspartate and alanine aminotransferases, gamma-glutamyltranspeptidase, blood count, C-reactive protein, tumor necrosis factor alpha.

Blood count and blood pressure - where are these results?

Why a placebo group was not used? Why the control group was not supplemented?

As the control group did not receive supplement, I would like to suggest a comparison of the post treatment parameters (T6) with control values (maybe some significant improvements could arise from it, if it happens, figure 1 could be added to n table 1 as T6)

Figure 3 does not show a stratification of Nafld patients. To show that, it would be necessary to establish subgroups separated by their BPA levels.

Figure 4 b should be rearranged in two separated figures.

Minor comments:

“Alimentary test” is an unusual expression, I would suggest a “food consumption assessment”.

Electronic program = software?

Line 120 instead of “elaborates” I suggest “evaluates”.

Line 121 elements = nutrients?

Table 1: P value for age; “average daily energy intake” would be more appropriated; also: “Plasma BPA”,  “Urine free BPA”, “Urine total BPA”, ”Plasma beta estradiol”.

Line 295 instead of “prescribe any behavioral rule” I would suggest “advised any behavioral change”

The “simplistic association” is not enough to support that no difference was observed in the daily energy intake. Another aspect that could be considered is the underreporting of food consumption by obese individuals (PMCID: PMC2550989)

Line 326 needs English reviewing.

341 – 344: in opposition to conclusions, the authors stated that the treatment was effective to reduce oxidative stress in Nafld patients.

Author Response

Response to reviewer 1: we thank the Reviewer for her/his comments.

Major comments:

  1. We agree with the observation of the reviewer. Actually, the indicated sentences in the conclusion section are confusing because they give to the reader the idea not to recommend the use of silybin in this clinical context despite the results obtained in the present study on metabolic, biochemical, hormonal and oxidative stress-related parameters. In the indicated sentences of the conclusion section we would have wanted to highlight that the use of silybin in this context is also useful to increase the BPA excretion rate but, surely, we tried to explain it wrongly. We modified the sentences according to the endpoint of the study.
  2. We added in the method section the methodology used.
  3. We added the requested results and the p-value in figure 1.
  4. The baseline comparison between NAFLD and control group was showed in table 1 and discussed abundantly in the discussion section (this comparison was useful in order to highlight the main differences in sense of BPA exposition between healthy patients and NAFLD ones). On the contrary, to assess the effect of therapy we made a comparison between two dependent groups: baseline and end of the treatment. The comparison was sustained by the statistical analysis, sample size and significance level. We decided to perform this study design in order to not treat control patients with a specific NAFLD therapy.
  5. We performed the analysis as requested. However, as expected, we didn’t obtain, from the comparison analysis of the T6 NAFLD group with the control group, statistically significant differences despite the reduction of several evaluated indices such as ALT, insulinemia, HOMA, CRP, etc (that anyway resulted not statistically significant increased in comparison to healthy subjects, obviously because of the NAFLD group has the disease). Specifically: insulin: p=0.472; glycemia: p=0.259; HOMA: p=0.612; GGT: p=0.760; AST: p=0.223; ALT: p=0.718; TNF: p=0.974; CRP: p=0.394; BPA: p=0.06; TBARS: p=0.338.
  6. We agree with the reviewer comment. “we stratified the NAFLD patients” is not the correct sentence. We modified it in the main text as suggested in order to indicate that, despite we didn’t construct class levels (transforming the continuous variables “TBARS levels” in an ordinal one) we found a direct proportionality relationship between thiobarbituric acid reactive substances (TBARS) and serum bisphenol A (BPA) levels in non-alcoholic fatty liver disease patients. We performed a multivariate analysis in order to evaluate the role of some confounding variables (ec. age, diabetes mellitus, stage of the disease) on the comparison between serum TBARS and BPA levels in NAFLD patients and this it is independent of the construction of subgroups or not.
  7. We modified the figure as suggested

Minor comments:

  1. We agree with the suggestion and changed the expression as suggested;
  2. We agree with the suggestion and changed the expression as suggested;
  3. We changed it as suggested;
  4. Yes, nutrients. We changed it as suggested;
  5. We added the age P-value and corrected the words as suggested;
  6. We changed it as suggested;
  7. We modified the sentence according to the suggestion and added the specific reference;
  8. We revised the English style supported by a mother tongue;
  9. In the submitted version of the manuscript we wanted to highlight that, independently of the antioxidant power of silybin, a simple BPA-free diet, as demonstrated in our previous paper, is able to reduce the BPA plasmatic amount. Here we demonstrated that the use of silybin is able to improve the rate of BPA excretion. Anyway, we agree with the comment and modified the sentences in the conclusion section according to the general sense of the paper.

Reviewer 2 Report

This study investigate in a clinical model and focuse on the potential role of BPA induced oxidative stress in NAFLD patients. The use of Realsil 100D could reduce TBARS and other metabolic parameters, defending the integrity of biological estrogen-dependent pathways. It is an interesting and pragmatic study, that given a strategy diet and treatment in NAFLD patients. I would like to seek some clarifications on the following issues:

  1. They were not clearly to present the results in Figure 2 and 4. It could be better to put the parameters individually, not pooled together in one figure. Moreover, the authors compared the biochemical parameters between baseline and after six-month of treatment in NAFLD patients. Could you also give the results with control group comparison? We can know the effect of the treatment.
  2. In Figure 5, the authors should descript the schematic representation illustration in detail.
  3. This study only assessed the male with NAFLD, which excluded the enrollment female patients. It may have a gender bias. The authors should clearly state in the title and abstract. Next may elevate the correlation of estrogen and the effect of BPA and Realsil 100D in female with NAFLD.

Author Response

Response to reviewer 2: we thank the Reviewer for her/his comments.

  1. We agree with the reviewer regarding the figure redraft and for this reason we divided the results of figure 2 into some subgraphs and the results of figure 4 in two different figures. As specified in the study design flow chart presented in figure 1 the control group didn’t follow a therapy. The baseline comparison between NAFLD and control group was showed in table 1 and discussed abundantly in the discussion section (this comparison was useful in order to highlight the main differences in sense of BPA exposition between healthy patients and NAFLD ones). On the contrary, to assess the effect of therapy we made a comparison between two dependent groups: baseline and the end of the treatment. The comparison was sustained by the statistical analysis, sample size and significance level. We also performed the analysis between T6 NAFLD group and controls. However, as expected, we didn’t obtain from the comparison analysis of the T6 NAFLD group with the control group statistically significant differences despite the reduction of several evaluated indices such as ALT, insulinemia, HOMA, CRP, etc (that anyway resulted not statistically significant increased in comparison to healthy subjects, obviously because NAFLD group has the disease).
  2. Sure, we agree with the suggestion. Thanks to this comment we noticed a mistake in the adaptation of the manuscript. We wrongly inserted the caption of the figure in the main text. We modified it as suggested.
  3. We specified in the title, abstract and main text this limitation. We also gave in the main text the motivation of this decision. It was very difficult to interpret the results in female subjects because of the physiologic change in oestrogen concentration due to the ovarian cycle. Anyway, in the future prospective (as added in the main text) we will design a specific gender study protocol.

Round 2

Reviewer 1 Report

The manuscript has potential, though it needs a detailed revision.

I would like to suggest to the authors to carefully read the whole manuscript to make several adjustments, as abbreviations (example CRP), brackets, prepositions, many English mistakes, unusual expression, also, formatting could be improved (as in statistical analysis, all the several paragraphs could be in just one). Moreover, some parts of the text are confusing.

Line 17 – no need of “Introduction”.

Line 25: “the results obtained were compared with those of healthy controls subjects”

Lines 65-67 – it is difficult to understand, moreover, please, clarify “but not the urine amount”

Line 72 – no need of “statistically significant”

Line 92 – “consecutive patients” would mean that were enrolled 32 patients in a row, without any patients matching the exclusion criteria (like female patients).

Line 103 – “we decided to not include” would be more appropriate.

Which technique was used to measure insulin?

Seems that R&D SYSTEMS a biotechnic brand Quantikine ELISA change to: (Quantikine ELISA kit, R&D SYSTEMS).

Line 131 – which software?

Was C-reactive protein included in the multivariate analysis? It is necessary to clearly indicate which parameters were included in this analysis and better describe it (lines 203-205).

Line 275 – “studied” still not appropriated – may be “we compared plasma TBARS and plasma BPA of patients with NAFLD, as can been seen in Figure 3A”.

Figure 2 – please add the y axis label [ex: Glucose (mg/dl)]. The x axis could only have “NAFLD - T0” and “NAFLD - T6”. Figure 4 and 5 as well.

Where are the results of estrogen levels after sylibin supplementation?

Lines 320 – 321 – need English revision.

Line 332 – consumption, not assumption. There is a lot of these little mistakes (I recommend to have the manuscript revised by a native English speaker).

I would like to recommend to not include p values and data in the discussion. Please, keep data in the results (frequency of fast-food consumption should be in Results).

Lines 333 to 335 – could be rewritten.

Figure 6 brings elements that were not discussed in the text, as hsp90. The figure could be improved as no clear pathway is showed.

Still, conclusion could be better written. Lines 453-459 are not necessary (at any point, Precise Medicine was discussed)  

Still, Conclusion should bring a clear idea of the effects obtained and answer the main objective [“to test in a clinical model, the effects of the administration of silybin with phospholipids and vitamin D (RealSIL 100D) in NAFLD patients exposed to BPA on metabolic, hormonal and oxidative stress-related parameters.” Lines 85-87].

Author Response

We thank the Reviewer for her/his comments.

  1. We did it as suggested.
  2. We did it as suggested.
  3. We modified the sentences.
  4. We did it as suggested.
  5. We enrolled 32 consecutive MALE patients. We modified the sentence.
  6. We did it as suggested.
  7. We used the common commercially available ELISA kit after ≥ 10 hours fast.
  8. We did it as suggested.
  9. Winfood Software 2.0 package (Medimatica s.r.l., Martinsicuro, Italy). We added it in the main text.
  10. No, the CRP was not included in the multivariate analysis because of it did not influence the relationship between TBARS and BPA. As dependent variable we used: age, BMI, waist/hip ratio, comorbidity and histologic liver staging. We decided to use those confounding variables because of the body composition could influence the BPA serum levels due to the lipo-affinity of this environmental pollutant. Moreover, we thought about the presence of comorbidities (obesity, hypertension, compensated cardiovascular diseases) and the histologic liver staging also could affect the relationship TBARS/BPA. We specified it better in the main text.
  11. We did it as suggested.
  12. We did it as suggested.
  13. T0=0.74±71 (ng/ml); T6=4.57±10.29 (ng/ml). We added the data in the specific section.
  14. We drew a mother tongue professional’s attention to the English review.
  15. We drew a mother tongue professional’s attention to the English review.
  16. We did it as suggested.
  17. We drew a mother tongue professional’s attention to the English review.
  18. We modified the figure 6 and complete the caption giving a complete explanation of the showed biological process.
  19. We modified the conclusion section bearing in mind the primary endpoint of the study.
  20. We modified the conclusion section bearing in mind the primary endpoint of the study.
